# Effects of changes in physical and sedentary behaviors on mental health and life satisfaction during the COVID-19 pandemic: Evidence from China

Xi Chen[1]*, Haiyan Gao[2], Binbin Shu[3], Yuchun Zou[4]

1 The Jockey Club School of Public Health and Primary Care, The Chinese University of Hong Kong, Hong Kong, Hong Kong, 2 Faculty of Humanities and Social Sciences, Beijing University of Technology, Chaoyang, China, 3 Department of Sociology, Shenzhen University, Shenzhen, China, 4 Institute of Sociology, Chinese Academy of Social Sciences, Beijing, China

* celiaxichen@cuhk.edu.hk

## Abstract

### Background

While restriction measures are critical in containing the COVID-19 outbreak, limited studies have investigated the behavioral and psychological impact of these measures. This study aimed to investigate the effects of physical and sedentary behavioral changes and online behavior during the COVID-19 pandemic on mental health and life satisfaction among the Chinese population.

### Methods

The data were obtained from a cross-sectional survey of 2145 residents aged between 18 and 80 in Hubei province, China between March 23, 2020, and April 9, 2020.

### Results

Participants who had high frequencies of physical activities before or during the COVID-19 outbreak exhibited higher levels of life satisfaction. Participants who increased their sitting time during the pandemic or kept sitting for more than eight hours before and during the pandemic reported worse mental health than those who maintained less sedentary behavior. Besides, participants who used the Internet for information seeking, communication, and entertainment more frequently reported better mental health and life satisfaction. In contrast, there was a positive association between commercial use of the Internet and symptoms of mental disorders.

### Conclusion

Given the link between physical and sedentary behavioral changes with worse mental well-being, strategies to reduce sedentariness and increase physical activity during the COVID-19 pandemic are needed.

**Data Availability Statement:** Data cannot be shared publicly because of data protection regulation. Data are available from the Chinese Social Quality Data Archive for researchers who

meet the criteria for access to confidential data. The data are available for research upon reasonable request and with permission from the Chinese Social Quality Data Archive: http://csqr.cass.cn/index.jsp.

**Funding:** This work was supported by the National Social Science Fund of China (grant no. 16ZDA079). The funders had no role in study design, data collection and analysis, decision to publish, or preparation of the manuscript.

**Competing interests:** The authors have declared that no competing interests exist.

## Introduction

### Mental health during the COVID-19 pandemic among the general population in China

The COVID-19 pandemic has heavily impacted many societies and economies. The total confirmed infection cases had reached over 483 million, and the death toll hit more than 6.1 million by March 30, 2022 [1]. To control the spreading of COVID-19, governments worldwide have implemented various restriction measures ranging from social distancing to societal shutdown [2]. While these measures are critical in containing the outbreak, they may compromise individuals' mental health and health activities [2, 3].

The outbreak of COVID-19 was first reported in Wuhan, the capital city of Hubei province, China, in December 2019. Authorities put Wuhan and 16 neighboring cities in Hubei under lockdown starting January 23, 2020. During the following weeks, COVID-19 cases increased rapidly and overwhelmed the health care system, especially in Wuhan. The situation improved when the Chinese government built massive temporary medical facilities to house COVID patients and transferred a large number of medical personnel to Hubei. The lockdowns were relaxed since March 23, 2020, and on April 8, 2020, the lockdown in Wuhan (the hardest-hit city in China) was officially lifted. Given the unprecedented social distancing measures, numerous studies have reported a high prevalence of mental health problems among Chinese adults during the COVID-19 pandemic [4–6]. However, most prior studies focused on socio-demographic backgrounds, risk perceptions, and attitudes toward COVID-19 as risk factors for psychological distress [7, 8]. Limited studies have investigated the effect of pandemic-related behavioral changes on population mental health.

### Mental health consequences of physical activity, sedentary behavior, and online activities

Numerous studies have shown that the COVID-19-related restriction measures disrupted daily routines and impacted lifestyle activities, including participation in sports and physical activity (PA) [9, 10]. The decline in PA was accompanied by increased sedentary (sitting) behavior [11]. Given the consistent benefits of regular PA for mental health [12–14], reductions in PA are likely to exacerbate COVID-19-fueled psychological distress. A review of 21 studies about the effect of PA on mental health during the COVID-19 pandemic suggested that people who performed PA regularly with high volume and frequency and kept the PA routines stable would experience fewer symptoms of anxiety and depression [15].

Additionally, accumulating evidence has revealed that, independent of PA levels, sedentary behavior is negatively associated with mental health [16, 17]. Several longitudinal investigations have shown that self-reported sedentary time predicts future mental health and wellbeing [18, 19]. A longitudinal study of populations in France and Switzerland found that sedentary behavior increased during the lockdown period of COVID-19, which was associated with poorer physical health, mental health, and subjective vitality [20]. Increasing sitting time during the COVID-19 pandemic may be a risk factor for mental health and life satisfaction.

Moreover, COVID-19-related measures, such as closing down schools, workplaces, and entertainment outlets coupled with self-quarantine, have led to an unprecedented level of Internet and digital technology use. People move almost every aspect of their daily activities to cyberspace at an accelerating speed, as a "new normal" in the COVID-19 era. However, it remains unclear whether and how Internet usage would exacerbate or alleviate COVID-19-fueled psychological distress. On the one hand, the Internet could enhance mental health by providing access to the latest information and enabling people to work, study, and shop

online that regularizes their routines during quarantines. On the other hand, excessive engagement in specific online activities such as gambling, gaming, social media use, and shopping may result in severe problems and elevate the risk of disordered or addictive use of the Internet [21, 22]. Given the mixed evidence regarding the effects of Internet usage on mental health, more studies are warranted to investigate the impact of Internet usage on mental wellbeing during the COVID-19 pandemic.

To date, limited studies have examined how changes in PA, sedentary behavior, and online behavior relate to mental health and quality of life among the Chinese population during the COVID-19 pandemic. A study of 4,898 adolescents in China showed that higher PA levels were significantly related to lower negative mood scores and higher positive mood scores during the COVID-19 pandemic [23]. More engagement in physical activities was also associated with reduced anxiety and depression among Chinese college students after adjusting for confounding demographic factors [24, 25]. While providing important insights, these studies often focused on subpopulations (e.g., adolescents, college students), and thus their findings cannot be generalized to the general population. A study of the Chinese adult population found that compared with that before the lockdown, their time engaged in daily PA decreased and sedentary time increased, and PA is positively associated with life satisfaction [26]. However, the authors did not examine the effect of COVID-19-induced behavioral changes on adverse mental health outcomes. Although some studies have found that PA participation was positively associated with mental health among the Chinese adults during the COVID-19 pandemic [27, 28], they did not specifically examine how the changes in PA and sitting time before and during the COVID-19 pandemic may affect mental health. Moreover, virtually no research has investigated how different types of online behavior during the pandemic may affect the Chinese population's psychological wellbeing during the COVID-19 pandemic.

### The present study

Given the lack of research on the effects of COVID-19-induced behavioral changes on mental wellbeing among the Chinese population, this study aims to investigate the impacts of behavioral changes in PA, sedentary behavior, and online behavior on individual psychological wellbeing and life satisfaction among residents in Hubei province, China, the original epicenter of the pandemic. The data collection was carried out between March 23, 2020, and April 9, 2020, while Hubei's lockdowns were gradually eased. Such data creates a unique opportunity for us to investigate Hubei residents' behavioral changes and psychological wellbeing during the pandemic. The findings of this study may improve our understanding and inform public health policies regarding the pattern and consequences of behavioral changes during the COVID-19 pandemic. Based on previous literature, we hypothesized that (1) the COVID-19 pandemic may be associated with changes in PA and sedentary behavior; (2) a decrease in PA may be negatively associated with mental health and life satisfaction; (3) an increase in sedentary behavior may have a negative effect on mental health and life satisfaction; and (4) Internet use may be positively associated with mental wellbeing and life satisfaction during the COVID-19 pandemic. In this study, we examined the impact of different types of Internet use, including information seeking, work/study, buying products, entertainment, reading/online learning, communication, and investment/financial management.

### Methods

#### Data

The data came from the "Public Attitude toward The Novel Coronavirus Epidemic in Hubei Province" survey, which was carried out by the China Academy of Science and Technology

Development Strategy, the Social Policy Research Institute at Renmin University, and the Institute of Sociology of the Chinese Academy of Social Sciences. The survey was conducted between March 23, 2020, and April 9, 2020. Authorities imposed a strict lockdown policy for cities in Hubei since January 23, 2020, and started to ease the lockdown since March 23, 2020, with the lockdown in Wuhan officially lifted on April 8, 2020.

The survey targeted all the residents aged between 18 and 80 in the urban and rural areas of Hubei. It was a combined online and phone survey. The online survey was conducted on a professional survey platform in China. The platform sent a notification to respondents in their sample bank with a link to allow access to the questionnaire. Only Hubei residents (with IP address locations in Hubei) could answer the questionnaire. No incentives were provided to the participants. A supplemented phone survey was carried out by trained research assistants to reach populations with limited access to the Internet (e.g., aged over 65, rural population). There were minimal differences between the two modes of data collection. Our research staff was asked not to provide any additional information but simply read the questions to the participants. Participants who did the online survey provided written consent. Participants who answered the questions via phone gave oral consent, and the interviewer signed a form pledging that he/she had gone through the proper procedures to obtain the verbal informed consent of the participant. The study was approved by the ethical committee of the Chinese Academy of Social Science, and all procedures were in accordance with the 1964 Helsinki declaration and its later amendments or comparable ethical standards. In total, 2355 participants completed the online or phone surveys. After deleting cases with at least one missing value, the final sample included 2145 respondents, of which 1944 (90.63%) completed the online survey, and 201 (9.37%) completed the phone survey.

## Measurements

*Symptoms of mental disorders* were measured by the 12-item Chinese Health Questionnaire scale (CHQ-12), which is widely used for screening non-psychotic mental health problems in the general population. Developed by Cheng and Williams (1986), CHQ was adapted from the General Health Questionnaire [29] with the inclusion of culturally relevant items in Chinese societies. CHQ-12 has been validated in the general population in mainland China [30, 31], Taiwan [32, 33], and Chinese people in foreign countries [34, 35]. Ten items of CHQ-12 were negatively worded (e.g., "Been suffering from headache or pressure in your head") and two items were positively worded (e.g., "Have you been getting along well with your family or friends recently?"). The responses for all items ranged from 0 = "not at all" to 3 = "very often." The responses for positive ones were reversely coded so that higher values indicated higher levels of mental disorder. The internal validity of the scale was good (Chronbach's alpha = 0.78). In this study, the cut-off point against the diagnosis of non-psychotic disorders was 13/14. About 32.07% of the participants scored greater than or equal to 14 and could be regarded as having symptoms consistent with potential mental disorders.

*Life satisfaction* was assessed by the question, "In general, how satisfied are you with your life?" with a 5-point scale from 1 = "very dissatisfied" to 5 = "very satisfied." Many studies and large-scale surveys have used such a single-item life satisfaction measure [36, 37]. Prior research has confirmed the reliability of this single-item measure of life satisfaction by showing that it performed very similarly compared to the multiple-item Satisfaction with Life Scale (SWLS) and correlated with external variables (e.g., health and affect) to a similar degree [38, 39].

**Physical activity.**  Participants reported the frequency of doing moderate-to-vigorous physical activity (MVPA) in 2019 and in the past month, followed by a brief explanation and

some examples of MVPA. MVPA refers to exercises such as running, playing ball, swimming, dancing, brisk walking, etc., for ≥10 minutes, which cause breathing and heart rate to increase. The response categories ranged from "never" to "every day." Exercise training guidelines for a quarantine situation due to COVID-19 suggested that healthy or asymptomatic persons should do aerobic exercises for at least three sessions per week (four or five seem optimal) [40]. Participants were categorized as "maintaining high PA" if they did PA > two times a week, as "decreased PA" if they did PA for > two times a week before the COVID-19 pandemic but did not do so afterward, as "increased PA" if they increased PA to > two times a week during the pandemic, and as "maintaining low PA" if they did PA for less than two times a week before and during the pandemic.

**Sitting time.** Participants reported average daily time spent sitting before (i.e., in early January 2020) and during the COVID-19 pandemic. Previous studies suggested that the association of daily sitting time and most long-term health outcomes is not linear, with a weekday sitting time below 8 hours/day associated with better perceived mental health and quality of life [41]. Participants were then classified as "maintaining low sitting time" if they reported ≤8h/day of sitting at both time points, as "increasing sitting time" if they reported ≤ 8h/day of sitting before the pandemic but reported > 8h/day of sitting after the pandemic, as "decreasing sitting time" if they reported > 8h/day of sitting before the pandemic but reported ≤ 8h/day of sitting after the pandemic, and as "maintaining high sitting time" if they reported > 8h/day of sitting at both time points.

**Online behavior.** Seven main and conceptually different types of Internet uses were assessed, including general information seeking; work; buying products (e.g., groceries); entertainment (e.g., playing online games, watching videos, listening to music); reading books/online class; communication with family and friends via online communication tools (e.g., WeChat); and investment and financial management. The responses ranged from 0 = "never" to 4 = "everyday". Higher values indicated a higher frequency of using the Internet for the corresponding activity.

In addition, a series of sociodemographic variables were measured, including age (continuous), sex (male vs. female), education (middle school or below, high school, college or above), occupation (managerial/professional position, manual/service/part-time worker, or other), China Communist Party membership (Party member vs. non-Party member), monthly personal income (from no income to >8000 *yuan*), area (rural vs. urban), city (Wuhan vs. non-Wuhan).

## Analysis

All the analyses were performed using Stata 16.0. Descriptive statistics were used to characterize the study population. Considering the variable of potential mental disorder was binary, we fit multiple logistic regression models to study the associations between changes in sitting time and PA during the COVID-19 pandemic and online behavior with symptoms of potential mental disorders. For the outcome variable of life satisfaction, we used OLS regression models to examine the effect of sedentary activity, PA, and online behavior after adjusting for sociodemographic variables. Variance inflation factors (VIFs) of the independent variables were estimated to check whether multicollinearity exists in the models. All the VIFs were below 2, suggesting that multicollinearity was not a significant concern.

## Results

Table 1 displays the background characteristics of the respondents.

**Table 1. Descriptive statistics of background variables.**

| Variable | N | (%) |
|---|---|---|
| Age, Mean (SD) | 33.70 (12.80) | |
| Sex | | |
| Male | 1045 | (48.72) |
| Female | 1100 | (51.28) |
| Education | | |
| Middle school or below | 257 | (11.98) |
| High school | 483 | (22.52) |
| College or above | 1405 | (65.50) |
| Occupation | | |
| Managerial/professional position | 495 | (23.08) |
| Manual/service worker | 1282 | (59.77) |
| Other | 368 | (17.16) |
| Communist Party membership | | |
| Party member | 343 | (15.99) |
| Non-Party member | 1802 | (84.01) |
| Monthly income | | |
| No income | 390 | (18.18) |
| $\leq$2000 | 451 | (21.03) |
| 2001–4000 | 642 | (29.93) |
| 4001–6000 | 394 | (18.37) |
| 6001–8000 | 159 | (7.41) |
| >8001 | 108 | (5.08) |
| Area | | |
| Rural | 165 | (7.69) |
| Urban | 1980 | (92.31) |
| City | | |
| Wuhan | 541 | (25.22) |
| Non-Wuhan | 1604 | (74.78) |

Table 2 shows the descriptive statistics of independent and dependent variables. Compared to the pre-COVID-19 period, about 4.66% of participants increased their sitting time, and approximately 15.10% reduced their PA during the pandemic. As for various online behavior, participants most frequently used the Internet for information seeking, followed by communication, entertainment, reading, and attending the online class, work and study. Relatively fewer participants used the Internet for commercial activities, including buying products and investment and financial management.

Table 3 presents the associations between physical activities and sedentary behavior changes, and various online behavior with depression (Models 1 and 2) and life satisfaction (Models 3 and 4) after adjusting for sociodemographic variables. As shown in Model 1, compared to participants who sit less than 8 hours before and after the pandemic, those who increased their sitting time during the pandemic and kept sitting for more than 8 hours a day at both time points were more likely to experience symptoms of mental disorders. However, changes in the frequency of PA were not associated with symptoms of mental disorders. Model 2 examined the impact of digital engagement and revealed that information seeking, entertainment, and communication activities were associated with lower levels of symptoms of mental disorders. Conversely, online commercial activities, including buying products and

**Table 2. Descriptive statistics of dependent and independent variables.**

| Variable | N (%) | Mean (SD) |
|---|---|---|
| Mental Health (CHQ-12) | | |
| Probable mental disorders | 688 (32.07) | |
| No probable mental disorders | 1457 (67.93) | |
| Life satisfaction | | 3.89 (0.94) |
| Physical activity | | |
| Maintained low | 229 (10.68) | |
| Increased | 117 (5.45) | |
| Decreased | 324 (15.10) | |
| Maintained high | 1475 (68.76) | |
| Sitting time | | |
| Maintained low | 1922 (89.60) | |
| Increased | 100 (4.66) | |
| Decreased | 53 (2.47) | |
| Maintained high | 70 (3.26) | |
| Online behavior | | |
| Information seeking | | 3.55 (0.80) |
| Work/study | | 2.91 (1.27) |
| Buying products | | 2.60 (1.06) |
| Entertainment | | 3.33 (0.91) |
| Reading/online learning | | 3.04 (1.17) |
| Communication | | 3.36 (0.94) |
| Investment/financial management | | 1.95 (1.35) |

doing investment online were positively associated with symptoms of mental disorders. Compared to those who participated in lower levels of physical activity, participants who increased, decreased, or maintained high frequencies of physical activity exhibited higher levels of life satisfaction. In contrast to the mixed effect of digital engagement on symptoms of mental disorders, most online behaviors were positively associated with life satisfaction. Specifically, using the Internet for information seeking, study/work, reading books/online class, and communication were positively associated with life satisfaction. Other online behaviors, such as buying products, entertainment, and investment/financial management, were not significantly associated with life satisfaction.

## Discussion

This study presents a timely investigation of the effects of behavioral changes (i.e., PA, sedentary behavior, and online behavior) induced by COVID-19-related restrictions on psychological wellbeing and life satisfaction among the general population in Hubei province, China, the original epicenter of the pandemic. With data obtained from 2145 Hubei residents during the pandemic, the findings showed (1) an expected decline in PA and an increase in sitting time during the pandemic; (2) an association between increased sitting time and symptoms of mental disorder; (3) higher life satisfaction for those doing more exercises; and (4) a mixed pattern of Internet uses and mental health and life satisfaction.

While it is well-documented that increased PA could reduce mental health problems [42–44], there is insufficient evidence regarding the effect of PA on positive mental health among the Chinese population. The findings of our study revealed that more involvement in PA

**Table 3. Adjusted associations between self-reported changes in behavior before and during the COVID-19 pandemic and mental health and life satisfaction.**

| | Probable mental disorders | | | | | | Life satisfaction | |
|---|---|---|---|---|---|---|---|---|
| | Model 1 | | Model 2 | | Model 3 | | Model 4 | |
| | OR | 95% CI | OR | 95% CI | b | 95% CI | b | 95% CI |
| Physical activity | | | | | | | | |
| Maintained low | Reference | | | | Reference | | | |
| Increased | 1.492 | [0.924,2.409] | | | 0.246* | [0.040,0.453] | | |
| Decreased | 0.721 | [0.488,1.065] | | | 0.238** | [0.081,0.395] | | |
| Maintained high | 1.322 | [0.966,1.808] | | | 0.425*** | [0.294,0.555] | | |
| Sitting time | | | | | | | | |
| Maintained low | Reference | | | | Reference | | | |
| Increased | 2.384*** | [1.579,3.600] | | | -0.010 | [-0.197,0.176] | | |
| Decreased | 1.031 | [0.564,1.884] | | | -0.015 | [-0.268,0.237] | | |
| Maintained high | 2.022** | [1.229,3.324] | | | -0.188 | [-0.411,0.035] | | |
| Online behavior | | | | | | | | |
| Information seeking | | | 0.666*** | [0.591,0.752] | | | 0.095*** | [0.043,0.148] |
| Work/study | | | 0.940 | [0.863,1.024] | | | 0.052** | [0.018,0.086] |
| Buying products | | | 1.217*** | [1.093,1.354] | | | 0.039 | [-0.004,0.081] |
| Entertainment | | | 0.878* | [0.784,0.984] | | | 0.012 | [-0.036,0.060] |
| Reading/online learning | | | 0.953 | [0.871,1.042] | | | 0.054** | [0.018,0.091] |
| Communication | | | 0.857** | [0.766,0.958] | | | 0.048* | [0.001,0.096] |
| Investment/financial management | | | 1.181*** | [1.091,1.277] | | | -0.006 | [-0.038,0.027] |
| Sex | | | | | | | | |
| Male | Reference | | Reference | | Reference | | Reference | |
| Female | 0.958 | [0.795,1.153] | 0.981 | [0.811,1.186] | -0.036 | [-0.115,0.043] | -0.039 | [-0.117,0.040] |
| Age | 1.004 | [0.996,1.011] | 1.003 | [0.995,1.011] | -0.001 | [-0.004,0.003] | 0.000 | [-0.003,0.003] |
| Education | | | | | | | | |
| Middle school or below | Reference | | Reference | | Reference | | Reference | |
| High school | 1.028 | [0.737,1.434] | 1.042 | [0.741,1.464] | 0.078 | [-0.063,0.220] | 0.077 | [-0.063,0.218] |
| College or above | 0.980 | [0.728,1.319] | 0.961 | [0.709,1.302] | 0.041 | [-0.085,0.167] | 0.031 | [-0.094,0.157] |
| Occupation | | | | | | | | |
| Managerial/professional position | Reference | | Reference | | Reference | | Reference | |
| Manual/service/part-time worker | 0.974 | [0.766,1.238] | 0.956 | [0.748,1.222] | 0.070 | [-0.031,0.172] | 0.086 | [-0.015,0.187] |
| Other | 1.133 | [0.836,1.536] | 1.165 | [0.852,1.593] | 0.095 | [-0.035,0.225] | 0.113 | [-0.016,0.242] |
| Party membership | | | | | | | | |
| Party member | Reference | | Reference | | Reference | | Reference | |
| Non-party member | 1.028 | [0.792,1.335] | 1.044 | [0.799,1.363] | 0.011 | [-0.100,0.121] | -0.004 | [-0.114,0.106] |
| Monthly income | | | | | | | | |
| No income | Reference | | Reference | | Reference | | Reference | |
| ≤2000 | 1.280 | [0.950,1.725] | 1.136 | [0.837,1.542] | -0.071 | [-0.197,0.055] | -0.064 | [-0.189,0.062] |
| 2001–4000 | 1.129 | [0.851,1.499] | 0.985 | [0.737,1.316] | 0.163** | [0.045,0.281] | 0.184** | [0.067,0.302] |
| 4001–6000 | 1.022 | [0.743,1.406] | 0.921 | [0.664,1.277] | 0.179** | [0.047,0.311] | 0.186** | [0.055,0.318] |
| 6001–8000 | 1.313 | [0.878,1.963] | 1.282 | [0.850,1.931] | 0.069 | [-0.103,0.241] | 0.069 | [-0.102,0.241] |
| >8001 | 1.783* | [1.132,2.808] | 1.480 | [0.930,2.355] | 0.312** | [0.113,0.511] | 0.326** | [0.128,0.524] |
| Area | | | | | | | | |
| Rural | Reference | | Reference | | Reference | | Reference | |
| Urban | 0.914 | [0.648,1.291] | 0.762 | [0.535,1.084] | 0.086 | [-0.061,0.234] | 0.103 | [-0.045,0.251] |
| City | | | | | | | | |
| Non-Wuhan | Reference | | Reference | | Reference | | Reference | |

(*Continued*)

**Table 3.** (Continued)

| | Probable mental disorders | | | | | | Life satisfaction | |
|---|---|---|---|---|---|---|---|---|
| | Model 1 | | Model 2 | | Model 3 | | Model 4 | |
| | OR | 95% CI | OR | 95% CI | b | 95% CI | b | 95% CI |
| Wuhan | 0.912 | [0.734,1.132] | 0.822 | [0.658,1.027] | -0.124** | [-0.215,-0.033] | -0.111* | [-0.201,-0.020] |
| Constant | 0.316 | [0.158,0.632] | 3.393 | [1.461,7.880] | 3.344*** | [3.051,3.637] | 2.684*** | [2.330,3.038] |
| Log lik. | -1319.53 | | -1273.05 | | -2851.45 | | -2837.18 | |

* $p < 0.05$,

** $p < 0.01$,

*** $p < 0.001$.

during the COVID-19 pandemic might effectively improve life satisfaction among Hubei residents. PA could be associated with life satisfaction through fitness- and health-related adaptations that enhance physical and mental health. PA may also have a revitalizing effect that strengthens self-control and supports goal pursuits [45, 46]. Considering we live in times of great psychological and emotional fragility, it is even more essential to promote PA that may help people adapt to the crisis and gain more life satisfaction. Our findings suggest that those who have low frequencies of PA during COVID-19 social distancing may need to be targeted with support for their quality of life. Policymakers and stakeholders should develop health education and communication that emphasize PA to improve life satisfaction during an infectious disease outbreak. As people have limited opportunities for doing PA outside their home during social distancing, future PA intervention to foster an Active and Healthy Confinement Lifestyle (AHCL) during a pandemic may adopt information and communications technology (ICT) solutions, such as home-based exercise games and fitness apps.

Additionally, our findings indicated that the overall time spent in sedentary behavior is positively associated with symptoms of mental disorders. Sedentary behavior could be associated with mental disorders via several possible pathways. For example, screen-based sedentary behaviors (e.g., TV viewing) are likely to induce addiction [47], internalizing symptoms [48], and low sleeping quality [49], potentially leading to heightened levels of mental distress. Further, engaging in sedentary behaviors may displace time spent in other activities such as household or work-related responsibilities or PA, which may be effective coping strategies for symptoms of mental disorders [50]. It is worth noting that specific sedentary behaviors may contribute to mental health in different ways. For example, a study found that specific screen-based recreation was associated with higher self-esteem among Australian adolescents in low-income communities [51]. Thus, future studies may investigate associations between different types of sedentary behavior and mental health.

Our findings showed that various Internet use types had different associations with mental health and life satisfaction during the COVID-19 pandemic among Hubei residents. To start, online information-seeking behavior was associated with a lower likelihood of mental disorders and increased life satisfaction. Such findings seemed inconsistent with other studies showing that exposure to COVID-19-related information may increase psychological distress [52, 53]. The discrepancy between our study and others may be that we measured the general information-seeking behavior rather than COVID-19 specific information seeking. Future studies may further investigate whether information-seeking behavior in general and about COVID-19 has differential effects on psychological wellbeing.

Also, online communication activity played an important role in improving mental health and life satisfaction, which appeared contradictory to prior studies showing that online communication (especially the use of social media) may deteriorate mental health due to its negative impact on face-to-face communications [54, 55]. Considering offline interactions were restricted by physical distancing and stay-at-home orders during the COVID-19 pandemic, online contact with friends and family may increase emotional support, and decrease loneliness and social isolation, which leads to lower symptoms of mental disorders and improved life satisfaction. Furthermore, our results have shown that online entertainment can reduce symptoms consistent with potential mental disorders. Using the Internet to relax may be a positive coping strategy to alleviate feelings of stress and isolation [56]. Similarly, working and learning remotely during the pandemic may keep people connected with their co-workers and classmates, despite physical isolation, and thus were positively associated with life satisfaction. Using the Internet for work and study may also help regularize daily routines, which is essential for buffering the adverse impact of stress exposure during a crisis [57].

While respondents who used the Internet for information seeking, communication, study/ work, reading books, and entertainment reported better mental health and life satisfaction, the commercial use of the Internet (e.g., shopping, investment) seemed to increase mental distress. We conjecture that the economic downturn and uncertainty caused by the pandemic may be why people who were involved in more online economic activities suffered more distress. Previous studies also found that online shopping may be a negative coping strategy for stress and was associated with increased symptoms of mental disorders [58, 59]. While online purchases were the dominant way for people to get groceries due to social distancing and stay-at-home orders, scholars have suggested setting a specific time and financial limits for online shopping to reduce the risk of disordered or addictive use [60].

Despite the significant findings, this study is not without limitations. First, due to the cross-sectional design employed in the present study, it cannot ascertain whether mental health problems lead to lower PA levels or whether lower levels of PA lead to mental health problems during the COVID-19 pandemic. Additional longitudinal studies are needed to infer the direction of the association. Second, our samples are not representative since the study was mainly conducted online. However, considering in-person surveys would not be appropriate or possible during the COVID-19 pandemic, online surveys are arguably the most feasible way to collect data in a timely matter. We also supplemented the online survey with a telephone survey to reach groups with limited access to the Internet. Future studies may try various sampling methods to gain a population-based representative sample. Third, participants were asked to self-report their PA level, thus introducing self-reporting bias into the findings. Moreover, we asked participants to retrospectively report their physical activity and sitting time prior to the COVID-19 pandemic, which may be subject to recall bias due to inaccurate memory. Future studies may use objective or direct measures of physical activity to increase precision and validate the self-report measures. Lastly, due to the online survey's length limitation, we only measured the PA frequency and the total amount of sedentary behavior during the pandemic. Future studies may further specify the level and type of PA and sedentary behavior and their relationship with mental health and life satisfaction.

## Conclusion

In conclusion, social distancing and quarantine measures to mitigate the COVID-19 pandemic may affect PA, mental health, and life satisfaction among the general population, with those experiencing a decrease in PA having lower life satisfaction, and those sitting more time reporting a higher likelihood of mental disorders. Additionally, the increasing use of the

Internet and digital technology throughout the pandemic also impacts individuals' mental health and life satisfaction. Our findings showed that participants who used the Internet for information seeking, communication, study/work, reading books, and entertainment more frequently reported better mental health and life satisfaction. In contrast, the commercial use of the Internet (e.g., shopping, investment) seemed to increase mental distress.

## Author Contributions

**Conceptualization:** Xi Chen, Binbin Shu.

**Data curation:** Xi Chen, Haiyan Gao, Yuchun Zou.

**Formal analysis:** Xi Chen, Binbin Shu.

**Methodology:** Xi Chen, Haiyan Gao.

**Validation:** Yuchun Zou.

**Writing – original draft:** Xi Chen, Haiyan Gao.

**Writing – review & editing:** Binbin Shu, Yuchun Zou.

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
