## [Decision Letter · Decision Letter 0]

24 Feb 2022

PONE-D-21-27357

Effects of changes in physical and sedentary behaviors on mental health and life satisfaction during the COVID-19 lockdown: Evidence from China

PLOS ONE

Dear Dr. Chen,

Thank you for submitting your manuscript to PLOS ONE.  After careful consideration, we feel that this manuscript has merit but does not fully meet PLOS ONE’s publication criteria as it currently stands. Therefore, we invite you to submit a revised version of the manuscript that addresses the points raised during the review process.

I apologize for the long wait you have had for feedback on your manuscript. One reviewer has submitted their comments and to avoid further delays, I will proceed with a decision and offer my comments, in addition to those of the reviewer. My decision is Minor Revision. The reviewer makes important suggestions. Please action these or explain why the suggestion was not actioned. In addition, please consider my comments as well. These include:

1. Please clarify the reasons for the restrictions on data availability. Stating “data available upon reasonable  request from the corresponding author” is not sufficient. Please explain your exceptional situation.

2. This is a cross-sectional study using survey methodology. Your limitations section (pages 13-14) does a very good job of conveying this material about the cross-sectional nature of the design. As you note, this type of design does not allow statements about one variable causing a change in another variable. For this reason, statements such as “the commercial use of the internet seemed to increase mental disorders” (in the abstract) should be phrased as “there was an association between commercial use of the internet and symptoms of mental disorders.” (Note comment 4 below that mental disorders per se were not assessed; rather, a self-report measure of symptoms of mental disorders was completed). Likewise, the statement “we hypothesized that (1) the lockdown may cause a decrease in PA and an increase in sedentary behavior” (page 6, as numbered) is not appropriate for this design. Please re-write as “….the lockdown may be associated with changes in PA and sedentary behavior.” More generally, throughout the writing, please edit to make sure the language used reflects the cross-sectional design and does not suggest causality or a longitudinal design.

3. Please clarify the consent process for the participants who went through a phone survey. Did participants who answered the questions via phone give oral consent, and those who did the online survey provide written consent?

4. The self-report measure used does not permit a true diagnosis of mental disorders. I recommend using language that indicates that the measure reports on symptoms consistent with a potential mental disorder.

5. What is meant by depression, page 10 as numbered? I don’t see a measure of depression. Is depression used as a synonym for “mental disorders”? Please harmonize the language. If the self-report measure is a measure of mental disorders, then the term mental disorders should be used to refer to findings from this measure.

6. Please make sure all the dates and details given in the text about the pandemic are correct. It is beyond the scope of the review process to verify this information.

7. Please make sure that your literature review is comprehensive and includes recent research on physical activity during the pandemic, such as this systematic review:

Wolf, S., Seiffer, B., Zeibig, J., Welkerling, J., Brokmeier, L., Atrott, B., Ehring, T., & Schuch, F. B. (2021). Is physical activity associated with less depression and anxiety during the COVID-19 pandemic? A rapid systematic review. Sports Medicine (Auckland), 51(8), 1771-1783. https://doi.org/10.1007/s40279-021-01468-z

Please submit your revised manuscript by April 7, 2022. If you will need more time than this to complete your revisions, please reply to this message or contact the journal office at plosone@plos.org. Please include the following items when submitting your revised manuscript:

We look forward to receiving your revised manuscript.

Kind regards,

Kristin Vickers, Ph.D.

Academic Editor

PLOS ONE

Journal Requirements:

2. Thank you for stating the following financial disclosure: "This work was supported by the National Social Science Fund of China (grant no. 16ZDA079)."

Reviewers' comments:

Reviewer's Responses to Questions

**Comments to the Author**

1. Is the manuscript technically sound, and do the data support the conclusions?

Reviewer #1: Yes

2. Has the statistical analysis been performed appropriately and rigorously? 

Reviewer #1: Yes

3. Have the authors made all data underlying the findings in their manuscript fully available?

Reviewer #1: No

4. Is the manuscript presented in an intelligible fashion and written in standard English?

Reviewer #1: Yes

5. Review Comments to the Author

Reviewer #1: PONE-D-21-27357

Abstract

The aim of the study is to examine physical and sedentary behaviour changes and online behavior on mental health and life satisfaction – but isn’t this regarding a specific time frame of lockdown during COVID (based on your title – “during the covid-19 lockdown”)? The abstract does not state that this study is specifically looking at participants behavioural and psychological outcomes during lockdown. Please clarify.

Please clarify “first suffered from the COVID-19 pandemic between March 23 and April 9, 2020” - the introduction says COVID-19 was first reported in Dec 2019 and locked since Jan 23, 2020 and then relaxed March 23 and April 8 – How did they first suffer in March? Do you mean data was collected from March 23-April 9 regarding their experience with the lockdown from Jan-Mar 2020? Please clarify what March 23-April 9 is regarding as the article is about a lockdown period and this time frame is when lockdown was relaxed.

Introduction

Page 4 – you mention how the internet could enhance mental health (access to information, enabling people to work, study and shop online) – what about the ability for people to connect with loved ones/friends via online platforms (e.g., zoom, skype, facetime), especially during a lockdown. This would allow people to still “see” and talk to others during a lockdown. This could also help mental health. Also, gaming isn’t necessarily a bad thing as people can game and chat with others which can be social, fun and something to do when in a lockdown. I’m not saying gaming could have negative effects, but it can also have good effects (specific to a lockdown).

Page 5 – “a study of 4,898 adolescents…” – this study is pre-COVID, correct?

The introduction doesn’t emphasize that you are looking at the lockdown period of COVID. The article mentions “during the COVID-19 pandemic”. Are you looking at the lockdown only or lockdown and non-lockdown periods during COVID? During a lockdown period, behavioural and psychological outcomes would be influenced more (can’t leave their homes) vs non-lockdown periods since people can leave their homes. Please specific what time frame you are looking at and why this time frame (lockdown only or both lockdown and non-lockdown periods, etc.). Also, make sure you clearly state this time frame throughout the manuscript. Also, this seems to be looking at the first lockdown which is also important to state.

“the data collection was carried out between March 23 and April 9, 2020, while Hubei’s lockdowns were gradually eased. Such data creates a unique opportunity for us to investigate Hubei residents’ behavioral changes and psychological wellbeing during the lockdown period.” – based on this sentence, my impression is you collected data during a non-lockdown period for participants to retrospectively report their behavioural and psychological outcomes during a lockdown (from Jan-Mar?), is this correct? You are specifically looking at lockdown effects. Why not collect data during the lockdown if its online anyways? Why wait until it eased when you are interested in lockdown effects? Please clarify.

Page 6 - Please specify the specific “online behaviors” you plan to examine in this study.

Methods

How did the participants get recruited? Any incentives for them to complete the survey?

Page 8 – for online behavior, what do you mean communication (talking with family members via zoom?). Please provide an example. Also, you looked at 7 internet uses which mostly seem like positive examples (based on your introduction) – you mention not having a hypothesis, but you only looked at internet uses that could enhance mental health; thus I would hypothesis internet use will have positive effects, correct? That said, why not include the negative internet uses you mentioned in the introduction such as gambling, social media use, shopping? Also, im assuming “buying products” is referring to “shop online that regularizes their routines”. Please provide an example of buying products.

Page 8 – Did you explain what “moderate to vigorous physical activity” means to the participant in the survey (e.g., moderate to vigorous physical activity means any activity that your heart rate increases and you are sweating) or leave it to the participant to interpret it? For example, someone may interpret physical activity as anything active (e.g., doing daily chores) within a day while someone else may interpret specific working out activities (e.g., running for an hour). As you can imagine, a daily chore may not be as “active” as someone who runs for an hour. You mention in the discussion that future studies should look at type of physical activity, which I agree is very important, but curious how much information was given to the participants regarding this question.

Page 8 – “information seeking” – is this specific COVID information seeking? Or other types of information seeking? How did you word this question?

Results

You have most of the information from Table 1 and 2 in text (e.g., “half of the respondents were female (51.28%).”). Thus, Tables 1 and 2 are redundant. I would put all the information in text and remove Tables 1 and 2. Please include the sample size for each variable (e.g., half of the respondents were female (n = X; 51.28%)). Or keep Table 1 and 2 but remove the text because it is redundant. If you keep the tables – add the sample size for each variable and separate the Mean (SD) data into one column and the % data into another column.

Page 10 - The “M = ” are missing for online class, work and study, buying products, investment and financial management.

Regression results – since you have table 3 with all the test statistics, you don’t need to have the statistics in text. You can remove the test statistic information and just explain the findings, as you have done (i.e., remove the numbers from this section that are in Table 3).

Most participants completed the survey alone online, but some participants completed via phone survey. Any differences between the style of data collection (e.g., participants could ask for clarification during a phone survey from the research assistant vs participants who completed the survey alone on a computer). Were research assistants asked to not provide any additional information but simply read the questions to the participants over the phone? How many completed the survey via phone vs online?

Discussion

Page 13 – I would not assume communication referring to use of social media, unless the question was framed this way. I would assume communication meaning communicating with others via phone or zoom/skype etc. When you clarify what “communication” means in the methods, this may make sense though.

I believe it is important to state that this study is looking at a lockdown period which is very different from normal day to day life (pre- and post-COVID). Something that was good during a lockdown (e.g., can’t work so they play video games all day for social interaction and something to do), may not be good when the lockdown is lifted (e.g., playing video games but should be at work). Thus, these findings are only relevant when a city/town are under lockdown.

Page 13 - please clarify what shopping means? Shopping for non-essentials (e.g., equipment) vs essentials (e.g., food)? Or does online shopping have a negative effect even when shopping for essentials (i.e., any type of shopping)? Though, I thought online shopping for essentials could give the impression of regularize daily routines which would be good for mental health? Please clarify.

Page 14 – another potential limitation – participants had to retrospectively report their physical activity and sitting time prior to COVID (especially 2019 MVPA rates) which could be inaccurate since it was based on memory (especially among older adults).

6. PLOS authors have the option to publish the peer review history of their article (what does this mean?). If published, this will include your full peer review and any attached files.

Reviewer #1: No

---

## [Author Response · Author response to Decision Letter 0]

31 Mar 2022

Response Letter

Editor's Comment

1.Please clarify the reasons for the restrictions on data availability. Stating "data available upon reasonable request from the corresponding author" is not sufficient. Please explain your exceptional situation.

Response: We have clarified the reasons for the restrictions on data availability, as shown below.

"Data cannot be shared publicly because of data protection regulation. Data are available from the Chinese Social Quality Data Archive for researchers who meet the criteria for access to confidential data. The data are available for research upon reasonable request and with permission from the Chinese Social Quality Data Archive: http://csqr.cass.cn/index.jsp"

2. This is a cross-sectional study using survey methodology. Your limitations section (pages 13-14) does a very good job of conveying this material about the cross-sectional nature of the design. As you note, this type of design does not allow statements about one variable causing a change in another variable. For this reason, statements such as "the commercial use of the internet seemed to increase mental disorders" (in the abstract) should be phrased as "there was an association between commercial use of the internet and symptoms of mental disorders." (Note comment 4 below that mental disorders per se were not assessed; rather, a self-report measure of symptoms of mental disorders was completed). Likewise, the statement "we hypothesized that (1) the lockdown may cause a decrease in PA and an increase in sedentary behavior" (page 6, as numbered) is not appropriate for this design. Please rewrite as "….the lockdown may be associated with changes in PA and sedentary behavior." More generally, throughout the writing, please edit to make sure the language used reflects the cross-sectional design and does not suggest causality or a longitudinal design.

Response: Thanks for your comment. We have revised the manuscript accordingly, using language that reflects the cross-sectional design. We detailed the changes below.

(1) We have rephrased the sentence "the commercial use of the internet seemed to increase mental disorders" (in the Abstract) as "there was a positive association between commercial use of the internet and symptoms of mental disorders."

(2) As the editor suggested, we have revised the first hypothesis as "the COVID-19 pandemic may be associated with changes in PA and sedentary behavior." (p.6)

(3) We have revised the second hypothesis, "a decrease in PA may deteriorate mental health and life satisfaction," as "a decrease in PA may be negatively associated with mental health and life satisfaction." (p.6)

3. Please clarify the consent process for the participants who went through a phone survey. Did participants who answered the questions via phone give oral consent, and those who did the online survey provide written consent?

Response: We have added information about the consent process in the revised manuscript (p.7).

"Participants who did the online survey provided written consent. Participants who answered the questions via phone gave oral consent, and the interviewer signed on a form pledging that he/she had gone through the proper procedures to obtain the verbal informed consent of the participant." 

4. The self-report measure used does not permit a true diagnosis of mental disorders. I recommend using language that indicates that the measure reports on symptoms consistent with a potential mental disorder.

Response: Thanks for the comment. We have revised the manuscript using language that indicates that the measure reports on symptoms consistent with a potential mental disorder. For example, 

(1) We rewrote the sentence "Mental health was measured by the 12-item Chinese Health Questionnaire scale (CHQ-12)" as "Symptoms of mental disorders were measured by the 12-item Chinese Health Questionnaire scale (CHQ-12)". (p.7)

(2) We have rephrased the sentence, "About 32.07% of the participants scored greater than or equal to 14 and could be regarded as having potential mental disorders." as "About 32.07% of the participants scored greater than or equal to 14 and could be regarded as having symptoms consistent with a potential mental disorder." (p.8)

(3) We have rephrased the sentence, "we fit multiple logistic regression models to study the associations between changes in sitting time and PA during the lockdown period of COVID-19 and online behavior with potential mental disorders." as "we fit multiple logistic regression models to study the associations between changes in sitting time and PA during the lockdown period of COVID-19 and online behavior with symptoms of potential mental disorders."

5. What is meant by depression, page 10 as numbered? I don't see a measure of depression. Is depression used as a synonym for "mental disorders"? Please harmonize the language. If the self-report measure is a measure of mental disorders, then the term mental disorders should be used to refer to findings from this measure.

Response: Thanks for the comment. We used depression as a synonym for mental disorders in the original manuscript. We have harmonized the language in the revised manuscript, particularly in the results and discussion parts. For example, we rewrote the sentence, "kept sitting for more than 8 hours a day at both time points were more likely to develop depressive symptoms" as "kept sitting for more than 8 hours a day at both time points were more likely to experience symptoms of mental disorders" (p.1q). 

We rewrote the sentence, "However, changes in the frequency of PA were not associated with depression" as "However, changes in the frequency of PA were not associated with symptoms of mental disorders" (p.11). Besides, we revised the sentence, "Previous studies also found that online shopping may be a negative coping strategy for stress and was associated with increased depression" as "Previous studies also found that online shopping may be a negative coping strategy for stress and was associated with increased symptoms of mental disorders." (p.13)

6. Please make sure all the dates and details given in the text about the pandemic are correct. It is beyond the scope of the review process to verify this information.

Response: We have double-checked the dates and details about the pandemic and confirmed that the details are correct.

7. Please make sure that your literature review is comprehensive and includes recent research on physical activity during the pandemic, such as this systematic review:

Wolf, S., Seiffer, B., Zeibig, J., Welkerling, J., Brokmeier, L., Atrott, B., Ehring, T., & Schuch, F. B. (2021). Is physical activity associated with less depression and anxiety during the COVID-19 pandemic? A rapid systematic review. Sports Medicine (Auckland), 51(8), 1771-1783. https://doi.org/10.1007/s40279-021-01468-z

Response: Thanks for the comment. We have conducted a more comprehensive literature review and added the reference suggested by the Editor and some more recent studies in China in the revised manuscript. The details of the changes are shown on pp. 4-5.

"A review of 21 studies about the effect of PA on mental health during the COVID-19 pandemic suggested that people who performed PA regularly with high volume and frequency and kept the PA routines stable would experience fewer symptoms of anxiety and depression [15]." (p.4)

"Although some studies have found that PA participation was positively associated with mental health among the Chinese adults during the COVID-19 pandemic [27,28], they did not specifically examine how the changes in PA and sitting time before and during the COVID-19 pandemic may affect mental health." (p.5)

 

Comments to the Author

Reviewer #1: PONE-D-21-27357

1. Abstract

The aim of the study is to examine physical and sedentary behaviour changes and online behavior on mental health and life satisfaction – but isn't this regarding a specific time frame of lockdown during COVID (based on your title – "during the covid-19 lockdown")? The abstract does not state that this study is specifically looking at participants behavioural and psychological outcomes during lockdown. Please clarify.

Response: Thanks for the comment. We have changed the title to "during the COVID-19 pandemic" since we share the same concern with the reviewer about the time frame of data collection. We have explained such a revision in response to the reviewer's comments #6 and #7. We stated explicitly that the study examined participants' behavioral and psychological outcomes during the COVID-19 pandemic in the revised Abstract.

"This study aimed to investigate the effects of physical and sedentary behavioral changes and online behavior during the COVID-19 pandemic on mental health and life satisfaction among the Chinese population."

2. Please clarify "first suffered from the COVID-19 pandemic between March 23 and April 9, 2020" - the introduction says COVID-19 was first reported in Dec 2019 and locked since January 23, 2020 and then relaxed March 23 and April 8 – How did they first suffer in March? Do you mean data was collected from March 23-April 9 regarding their experience with the lockdown from Jan-Mar 2020? Please clarify what March 23-April 9 is regarding as the article is about a lockdown period and this time frame is when lockdown was relaxed.

Response: Sorry for the confusion. We used "whose capital city Wuhan first suffered from the COVID-19 pandemic" to indicate that Hubei was the epicenter of the initial outbreak of the COVID-19 pandemic. To avoid confusion, we have deleted "whose capital city Wuhan first suffered from the COVID-19 pandemic" in the revised Abstract. The method section in the Abstract now reads:

"The data were obtained from a cross-sectional survey of 2145 residents aged between 18 and 80 in Hubei province, China between March 23, 2020, and April 9, 2020."

3. Introduction

Page 4 – you mention how the internet could enhance mental health (access to information, enabling people to work, study and shop online) – what about the ability for people to connect with loved ones/friends via online platforms (e.g., zoom, skype, facetime), especially during a lockdown. This would allow people to still "see" and talk to others during a lockdown. This could also help mental health. Also, gaming isn't necessarily a bad thing as people can game and chat with others which can be social, fun and something to do when in a lockdown. I'm not saying gaming could have negative effects, but it can also have good effects (specific to a lockdown).

Response: Thanks for the comment. We have included such a variable in our study. Our measure of online communication refers to the contact with family and friends via online communication tools (e.g., WeChat). WeChat is the most widely used communication tool in China, enabling people to see and talk to others. Also, we agree with the reviewer that online entertainment, such as gaming, could be a stress coping strategy and positively affect mental health during the pandemic. Our results actually support such an argument. The revised manuscript has included a hypothesis about Internet use and mental wellbeing. The corresponding changes are:

"(4) Internet use may be positively associated with mental wellbeing and life satisfaction during the COVID-19 pandemic." (p.6)

4. Page 5 – "a study of 4,898 adolescents…" – this study is pre-COVID, correct?

Response: That study was conducted during the COVID-19 pandemic. We have clarified it in the revised manuscript.

"A study of 4,898 adolescents in China showed that higher PA levels were significantly related to lower negative mood scores and higher positive mood scores during the COVID-19 pandemic" (p.5).

5. The introduction doesn't emphasize that you are looking at the lockdown period of COVID. The article mentions "during the COVID-19 pandemic". Are you looking at the lockdown only or lockdown and non-lockdown periods during COVID? During a lockdown period, behavioural and psychological outcomes would be influenced more (can't leave their homes) vs non-lockdown periods since people can leave their homes. Please specific what time frame you are looking at and why this time frame (lockdown only or both lockdown and non-lockdown periods, etc.). Also, make sure you clearly state this time frame throughout the manuscript. Also, this seems to be looking at the first lockdown which is also important to state.

Response: Thanks for the comment. We would like to clarify the timeline of the lockdown of Hubei province. Chinese authorities imposed a lockdown on Wuhan, the capital city of Hubei province, on January 23, 2020, and enforced similar measures in 16 neighboring cities in Hubei province, affecting approximately 57 million people (Xiong, 2020). The authorities relaxed the Hubei lockdown on March 23, 2020, and officially lifted the Wuhan lockdown on April 8, 2020, after no new deaths transpired for the first time (He, 2020). The study was conducted between March 23, 2020, and April 9, 2020, when the lockdowns in Hubei were relaxed and eventually lifted. We asked the participants to report their physical activity participation and sitting time in the past two weeks and the past month, respectively, which may be during the lockdown or non-lockdown periods. Thus, we thought it would be better to use "COVID-19 pandemic" instead of "COVID-19 lockdown". We have used such a term throughout the manuscript.

6. "the data collection was carried out between March 23 and April 9, 2020, while Hubei's lockdowns were gradually eased. Such data creates a unique opportunity for us to investigate Hubei residents' behavioral changes and psychological wellbeing during the lockdown period." – based on this sentence, my impression is you collected data during a non-lockdown period for participants to retrospectively report their behavioural and psychological outcomes during a lockdown (from Jan-Mar?), is this correct? You are specifically looking at lockdown effects. Why not collect data during the lockdown if its online anyways? Why wait until it eased when you are interested in lockdown effects? Please clarify.

Response: In the revised manuscript, we did not emphasize that we are looking at the lockdown period of the pandemic. We provided the reason in response to the reviewer's comment #5.

7. Page 6 - Please specify the specific "online behaviors" you plan to examine in this study.

Response: Thanks for the suggestion. We have specified the "online behaviors" in the revised manuscript (p.6). The corresponding changes are:

"Besides, we examined the impact of different types of Internet use, including information seeking, work/study, buying products, entertainment, reading/online learning, communication with family and friends, and investment/financial management."

8. Methods

How did the participants get recruited? Any incentives for them to complete the survey?

Response: We have provided more information about subject recruitment in the revised manuscript (p.7).

"The online survey was carried out on a professional survey platform in China. The platform sent a notification to respondents in their sample bank with a link to allow access to the questionnaire. Only Hubei residents (with IP address locations in Hubei) can answer the questionnaire. No incentives were provided to the participants."

9. Page 8 – for online behavior, what do you mean communication (talking with family members via zoom?). Please provide an example. Also, you looked at 7 internet uses which mostly seem like positive examples (based on your introduction) – you mention not having a hypothesis, but you only looked at internet uses that could enhance mental health; thus I would hypothesis internet use will have positive effects, correct? That said, why not include the negative internet uses you mentioned in the introduction such as gambling, social media use, shopping? Also, im assuming "buying products" is referring to "shop online that regularizes their routines". Please provide an example of buying products.

Response: Thanks for the questions. To clarify, communication in this study was measured by asking the respondents whether they communicated with family and friends via online communication tools (e.g., WeChat). WeChat is the most widely used communication app in China, which acted as critical infrastructure in fulfilling people's practical, emotional, and medical needs during the COVID lockdown in China (Qian & Hanser, 2020). We have clarified this in the revised manuscript as below.

"communication with family and friends via online communication tools (e.g., WeChat)" (p.9).

As the reviewer suggested, we have included an example for buying products in the revised manuscript as "buying products (e.g., groceries)" (p. 9).

10. Page 8 – Did you explain what "moderate to vigorous physical activity" means to the participant in the survey (e.g., moderate to vigorous physical activity means any activity that your heart rate increases and you are sweating) or leave it to the participant to interpret it? For example, someone may interpret physical activity as anything active (e.g., doing daily chores) within a day while someone else may interpret specific working out activities (e.g., running for an hour). As you can imagine, a daily chore may not be as "active" as someone who runs for an hour. You mention in the discussion that future studies should look at type of physical activity, which I agree is very important, but curious how much information was given to the participants regarding this question.

Response: We asked the participants about the frequency of doing moderate-to-vigorous physical activity (MVPA), followed by a brief explanation and some examples of MVPA. In the survey, MVPA refers to exercises such as running, playing ball, swimming, dancing, brisk walking, etc. for ≥10 minutes. Typically, physical activity needs to continue for at least 10 minutes to be considered a session of exercise. We have clarified the measure of "moderate to vigorous physical activity" in the revised manuscript as below:

"Participants reported the frequency of doing moderate-to-vigorous physical activity (MVPA) in 2019 and in the past month, followed by a brief explanation and some examples of MVPA. MVPA refers to exercises such as running, playing ball, swimming, dancing, brisk walking, etc., for ≥10 minutes, which cause breathing and heart rate to increase." (p.8)

11. Page 8 – "information seeking" – is this specific COVID information seeking? Or other types of information seeking? How did you word this question?

Response: Thanks for the question. The information seeking was measured by general information seeking behavior, not specific COVID-19 information seeking. We have clarified this in the revised manuscript (p.9).

"Seven main and conceptually different Internet uses were accessed, including general information seeking;"

12. Results

You have most of the information from Table 1 and 2 in text (e.g., "half of the respondents were female (51.28%)."). Thus, Tables 1 and 2 are redundant. I would put all the information in text and remove Tables 1 and 2. Please include the sample size for each variable (e.g., half of the respondents were female (n = X; 51.28%)). Or keep Table 1 and 2 but remove the text because it is redundant. If you keep the tables – add the sample size for each variable and separate the Mean (SD) data into one column and the % data into another column.

Response: Thanks for the suggestions. To be consistent with the reporting style in most studies, we decided to keep Tables 1 and 2. As the reviewer suggested, we have removed the statistics and only kept a very brief description of PA, sitting time and online behavior in the revised manuscript.

13. Page 10 - The "M =" are missing for online class, work and study, buying products, investment and financial management.

Response: Thanks for pointing this out. However, as the reviewer suggested in Comment #12, we have deleted these statistics as they are redundant.

14. Regression results – since you have table 3 with all the test statistics, you don't need to have the statistics in text. You can remove the test statistic information and just explain the findings, as you have done (i.e., remove the numbers from this section that are in Table 3).

Response: Thanks for the suggestion. We have removed the statistics in the text.

15. Most participants completed the survey alone online, but some participants completed via phone survey. Any differences between the style of data collection (e.g., participants could ask for clarification during a phone survey from the research assistant vs participants who completed the survey alone on a computer). Were research assistants asked to not provide any additional information but simply read the questions to the participants over the phone? How many completed the survey via phone vs online?

Response: Thanks for the questions. We have clarified the two modes of data collection in the revised manuscript as below:

"There are minimal differences between the two modes of data collection. Our research staff will be asked not to provide any additional information but simply read the questions to the participants." (p.7)

We also included the number of participants who completed the survey online or via telephone in the revised manuscript.

"After deleting cases with at least one missing value, the final sample included 2145 respondents, of which 1944 (90.63%) completed the online survey, and 201 (9.37%) completed the phone survey." (p.7)

16. Discussion

Page 13 – I would not assume communication referring to use of social media, unless the question was framed this way. I would assume communication meaning communicating with others via phone or zoom/skype etc. When you clarify what "communication" means in the methods, this may make sense though.

Response: Thanks for the question. We have clarified this measurement in the revised manuscript.

"communication with family and friends via online communication tools (e.g., WeChat)"

17. I believe it is important to state that this study is looking at a lockdown period which is very different from normal day to day life (pre- and post-COVID). Something that was good during a lockdown (e.g., can't work so they play video games all day for social interaction and something to do), may not be good when the lockdown is lifted (e.g., playing video games but should be at work). Thus, these findings are only relevant when a city/town are under lockdown.

Response: Thanks again for the comment. As we responded to the reviewer earlier, we would not emphasize that we are looking at the lockdown effects. We use "during the COVID-19 pandemic" throughout the manuscript.

18. Page 13 - please clarify what shopping means? Shopping for non-essentials (e.g., equipment) vs essentials (e.g., food)? Or does online shopping have a negative effect even when shopping for essentials (i.e., any type of shopping)? Though, I thought online shopping for essentials could give the impression of regularize daily routines which would be good for mental health? Please clarify.

Response: Thanks for the comment. We did not specify types of shopping in the survey. It could be shopping for essentials or non-essentials.

19. Page 14 – another potential limitation – participants had to retrospectively report their physical activity and sitting time prior to COVID (especially 2019 MVPA rates) which could be inaccurate since it was based on memory (especially among older adults).

Response: Thanks for pointing this out. We have added the reporting bias in the limitation of the revised manuscript. The corresponding revisions are shown below:

"Moreover, we asked participants to retrospectively report their physical activity and sitting time prior to the COVID-19 pandemic, which may be subject to recall bias due to inaccurate memory. Future studies may use objective or direct measures of physical activity to increase precision and validate the self-report measures." (p.15)

References:

He J. China's State Machinery Will Beat Coronavirus Crisis, but at What Cost? South China Morning Post, February 1, 2020.

Xiong, Y. All major cities in China's Hubei province under lockdown. 2020. Retrieved from. https://edition.cnn.com/asia/live-news/coronavirus-outbreak-02-02-20-intl-hnk/h_9d7932db226e2e5109eabc8e4f5ae6c9

Qian Y, Hanser A. How did Wuhan residents cope with a 76-day lockdown?. Chinese Sociological Review. 2021 January 1;53(1):55-86.

---

## [Decision Letter · Decision Letter 1]

4 May 2022

PONE-D-21-27357R1Effects of changes in physical and sedentary behaviors on mental health and life satisfaction during the COVID-19 pandemic: Evidence from ChinaPLOS ONE

Dear Dr. Chen,

Thank you for submitting your revised manuscript to PLOS ONE. Thank you as well for addressing the comments made by the reviewer and me. We appreciate your efforts. Unfortunately, the revised manuscript contains some minor issues and typos that need to be fixed. My decision is therefore a Minor Revision. We invite you to submit a revised version of the manuscript that addresses the edits that I delineate in the attachment (please look at the highlighted parts, and then the sticky notes for each highlight).

We look forward to receiving your revised manuscript.

Kind regards,

Kristin Vickers, Ph.D.

Academic Editor

PLOS ONE

Journal Requirements:

Reviewers' comments:

Reviewer's Responses to Questions

**Comments to the Author**

1. If the authors have adequately addressed your comments raised in a previous round of review and you feel that this manuscript is now acceptable for publication, you may indicate that here to bypass the “Comments to the Author” section, enter your conflict of interest statement in the “Confidential to Editor” section, and submit your "Accept" recommendation.

Reviewer #1: All comments have been addressed

2. Is the manuscript technically sound, and do the data support the conclusions?

Reviewer #1: Yes

3. Has the statistical analysis been performed appropriately and rigorously? 

Reviewer #1: Yes

4. Have the authors made all data underlying the findings in their manuscript fully available?

Reviewer #1: Yes

5. Is the manuscript presented in an intelligible fashion and written in standard English?

Reviewer #1: Yes

6. Review Comments to the Author

Reviewer #1: The authors have answered all of my questions and the manuscript looks good. I have accepted the manuscript for publication.

7. PLOS authors have the option to publish the peer review history of their article (what does this mean?). If published, this will include your full peer review and any attached files.

Reviewer #1: No

---

## [Author Response · Author response to Decision Letter 1]

5 May 2022

Dear Editor,

Thank you very much for your comments and suggestions on our manuscript. We have revised the manuscript accordingly. Please let us know if further revisions are needed.

Please note the the reviewer had no further comments on our previous revision, so we did not draft a response letter to reviewers.

---

## [Editor Report · Decision Letter 2]

18 May 2022

Effects of changes in physical and sedentary behaviors on mental health and life satisfaction during the COVID-19 pandemic: Evidence from China

PONE-D-21-27357R2

Dear Dr. Chen,

We’re pleased to inform you that your manuscript has been judged scientifically suitable for publication and will be formally accepted for publication once it meets all outstanding technical requirements.

Kind regards,

Kristin Vickers, Ph.D.

Academic Editor

PLOS ONE

Editor Comments: The authors have addressed all comments of the reviewer and the academic editor.

---

## [Editor Report · Acceptance letter]

10 Jun 2022

PONE-D-21-27357R2 

Effects of changes in physical and sedentary behaviors on mental health and life satisfaction during the COVID-19 pandemic: Evidence from China 

Dear Dr. Chen:

I'm pleased to inform you that your manuscript has been deemed suitable for publication in PLOS ONE. Congratulations! Your manuscript is now with our production department. 

Kind regards, 

on behalf of

Dr. Kristin Vickers 

Academic Editor

PLOS ONE